# Adversarial Multiple Source Domain Adaptation

**Han Zhao**[†*], **Shanghang Zhang**[†‡*], **Guanhang Wu**[†]
**João P. Costeira**[‡], **José M. F. Moura**[†], **Geoffrey J. Gordon**[†]
[†]Carnegie Mellon University      [‡]IST, Universidade de Lisboa
{hzhao1,shanghaz,guanhanw,moura,ggordon}@andrew.cmu.edu,
jpc@isr.ist.utl.pt

## Abstract

While domain adaptation has been actively researched, most algorithms focus on the single-source-single-target adaptation setting. In this paper we propose new generalization bounds and algorithms under both classification and regression settings for unsupervised multiple source domain adaptation. Our theoretical analysis naturally leads to an efficient learning strategy using adversarial neural networks: we show how to interpret it as learning feature representations that are invariant to the multiple domain shifts while still being discriminative for the learning task. To this end, we propose multisource domain adversarial networks (MDAN) that approach domain adaptation by optimizing task-adaptive generalization bounds. To demonstrate the effectiveness of MDAN, we conduct extensive experiments showing superior adaptation performance on both classification and regression problems: sentiment analysis, digit classification, and vehicle counting.

## 1   Introduction

The success of machine learning has been partially attributed to rich datasets with abundant annotations [40]. Unfortunately, collecting and annotating such large-scale training data is prohibitively expensive and time-consuming. To solve these limitations, different labeled datasets can be combined to build a larger one, or synthetic training data can be generated with explicit yet inexpensive annotations [41]. However, due to the possible shift between training and test samples, learning algorithms based on these cheaper datasets still suffer from high generalization error. Domain adaptation (DA) focuses on such problems by establishing knowledge transfer from a labeled source domain to an unlabeled target domain, and by exploring domain-invariant structures and representations to bridge the gap [38]. Both theoretical results [8, 22, 32, 33, 49] and algorithms [1, 2, 6, 19, 20, 23, 25, 26, 30, 39] for DA have been proposed. Most theoretical results and algorithms with respect to DA focus on the single-source-single-target setting [17, 31, 42, 45, 46]. However, in many application scenarios, the labeled data available may come from multiple domains with different distributions. As a result, naive application of the single-source-single-target DA algorithms may lead to suboptimal solutions. Such problem calls for an efficient technique for multiple source domain adaptation. Some existing multisource DA methods [16, 23, 24, 43, 51] cannot lead to effective deep learning based algorithms, leaving much space to be improved for their performance.

In this paper, we analyze the multiple source domain adaptation problem and propose an adversarial learning strategy based on our theoretical results. Specifically, we give new generalization bounds for both classification and regression problems under domain adaptation when there are multiple source domains with labeled instances and one target domain with unlabeled instances. Our theoretical results build on the seminal theoretical model for domain adaptation introduced by Blitzer et al. [9] and Ben-David et al. [8], where a divergence measure, known as the $\mathcal{H}$-divergence, was proposed to measure the distance between two distributions based on a given hypothesis space $\mathcal{H}$. Our new result generalizes the bound [8, Thm. 2] to the case when there are multiple source domains, and

---

[*]The first two authors contributed equally to this work.

to regression problems as well. The new bounds achieve a finite sample error rate of $\tilde{O}(\sqrt{1/km})$, where $k$ is the number of source domains and $m$ is the number of labeled training instances from each domain. We provide detailed comparisons with existing work in Section 3.

Interestingly, our bounds also lead to an efficient algorithm using adversarial neural networks. This algorithm learns both domain invariant and task discriminative features under multiple domains. Specifically, we propose a novel MDAN model by using neural networks as rich function approximators to instantiate the generalization bound we derive (Fig. 1). MDAN can be viewed as computationally efficient approximations to optimize the parameters of the networks in order to minimize the bounds. We introduce two versions of MDAN: The hard version optimizes directly a simple worst-case generalization bound, while the soft version leads to a more data-efficient model and optimizes an average case and task-adaptive bound. The optimization of MDAN is a minimax saddle point problem, which can be interpreted as a zero-sum game with two participants competing against each other to learn invariant features. MDAN combine feature extraction, domain classification, and task learning in one training process. We propose to use stochastic optimization with simultaneous updates to optimize the parameters in each iteration.

**Contributions**. Our contributions are three-fold: 1). Theoretically, we provide average case generalization bounds for both classification and regression problems under the multisource domain adaptation setting. 2). Inspired by our theoretical results, we also propose efficient algorithms that tackle multisource domain adaptation problems using adversarial learning strategy. 3). Empirically, to demonstrate the effectiveness of MDAN as well as the relevance of our theoretical results, we conduct extensive experiments on real-world datasets, including both natural language and vision tasks, classification and regression problems. We achieve consistently superior adaptation performances on all the tasks, validating the effectiveness of our models.

## 2  Preliminary

We first introduce the notations and review a theoretical model for domain adaptation when there is one source and one target [7–9, 27]. The key idea is the $\mathcal{H}$-divergence to measure the discrepancy between two distributions. Other theoretical models for DA exist [12, 13, 33, 35]; we choose to work with the above model because this distance measure has a particularly natural interpretation and can be well approximated using samples from both domains.

**Notations** We use *domain* to represent a distribution $\mathcal{D}$ on input space $\mathcal{X}$ and a labeling function $f : \mathcal{X} \to [0,1]$. In the setting of one source one target domain adaptation, we use $\langle \mathcal{D}_S, f_S \rangle$ and $\langle \mathcal{D}_T, f_T \rangle$ to denote the source and target, respectively. A *hypothesis* is a function $h : \mathcal{X} \to [0,1]$. The *error* of a hypothesis $h$ w.r.t. a labeling function $f$ under distribution $\mathcal{D}_S$ is defined as: $\varepsilon_S(h,f) := \mathbb{E}_{\mathbf{x} \sim \mathcal{D}_S}[|h(\mathbf{x}) - f(\mathbf{x})|]$. When $f$ and $h$ are binary classification functions, this definition reduces to the probability that $h$ disagrees with $f$ under $\mathcal{D}_S$: $\mathbb{E}_{\mathbf{x} \sim \mathcal{D}_S}[|h(\mathbf{x}) - f(\mathbf{x})|] = \mathbb{E}_{\mathbf{x} \sim \mathcal{D}_S}[\mathbb{I}(f(\mathbf{x}) \neq h(\mathbf{x}))] = \Pr_{\mathbf{x} \sim \mathcal{D}_S}(f(\mathbf{x}) \neq h(\mathbf{x}))$.

We define the *risk* of hypothesis $h$ as the error of $h$ w.r.t. a true labeling function under domain $\mathcal{D}_S$, i.e., $\varepsilon_S(h) := \varepsilon_S(h, f_S)$. As common notation in computational learning theory, we use $\widehat{\varepsilon}_S(h)$ to denote the empirical risk of $h$ on the source domain. Similarly, we use $\varepsilon_T(h)$ and $\widehat{\varepsilon}_T(h)$ to mean the true risk and the empirical risk on the target domain. $\mathcal{H}$-divergence is defined as follows:

**Definition 1.** Let $\mathcal{H}$ be a hypothesis class for instance space $\mathcal{X}$, and $\mathcal{A}_{\mathcal{H}}$ be the collection of subsets of $\mathcal{X}$ that are the support of some hypothesis in $\mathcal{H}$, i.e., $\mathcal{A}_{\mathcal{H}} := \{h^{-1}(\{1\}) \mid h \in \mathcal{H}\}$. The distance between two distributions $\mathcal{D}$ and $\mathcal{D}'$ based on $\mathcal{H}$ is: $d_{\mathcal{H}}(\mathcal{D}, \mathcal{D}') := 2\sup_{A \in \mathcal{A}_{\mathcal{H}}} |\Pr_{\mathcal{D}}(A) - \Pr_{\mathcal{D}'}(A)|$.

When the hypothesis class $\mathcal{H}$ contains all the possible measurable functions over $\mathcal{X}$, $d_{\mathcal{H}}(\mathcal{D}, \mathcal{D}')$ reduces to the familiar total variation. Given a hypothesis class $\mathcal{H}$, we define its symmetric difference w.r.t. itself as: $\mathcal{H}\Delta\mathcal{H} = \{h(\mathbf{x}) \oplus h'(\mathbf{x}) \mid h, h' \in \mathcal{H}\}$, where $\oplus$ is the XOR operation. Let $h^*$ be the optimal hypothesis that achieves the minimum combined risk on both the source and the target domains: $h^* := \arg\min_{h \in \mathcal{H}} \varepsilon_S(h) + \varepsilon_T(h)$, and use $\lambda$ to denote the combined risk of the optimal hypothesis $h^*$: $\lambda := \varepsilon_S(h^*) + \varepsilon_T(h^*)$. Ben-David et al. [7] and Blitzer et al. [9] proved the following generalization bound on the target risk in terms of the source risk and the discrepancy between the single source domain and the target domain:

**Theorem 1** ([9]). Let $\mathcal{H}$ be a hypothesis space of $VC$-dimension $d$ and $\widehat{\mathcal{D}}_S$ ($\widehat{\mathcal{D}}_T$) be the empirical distribution induced by sample of size $m$ drawn from $\mathcal{D}_S$ ($\mathcal{D}_T$). Then w.p.b. at least $1 - \delta$, $\forall h \in \mathcal{H}$,

$$\varepsilon_T(h) \leq \widehat{\varepsilon}_S(h) + \frac{1}{2}d_{\mathcal{H}\Delta\mathcal{H}}(\widehat{\mathcal{D}}_S, \widehat{\mathcal{D}}_T) + \lambda + O\left(\sqrt{\frac{d\log(m/d) + \log(1/\delta)}{m}}\right) \qquad (1)$$

The bound depends on $\lambda$, the optimal combined risk that can be achieved by hypothesis in $\mathcal{H}$. The intuition is if $\lambda$ is large, we cannot hope for a successful domain adaptation. One notable feature is that the empirical discrepancy distance between two samples can be approximated by a discriminator to distinguish instances from two domains.

## 3 Generalization Bound for Multiple Source Domain Adaptation

In this section we discuss two approaches to obtain generalization guarantees for multiple source domain adaptation in both classification and regression settings, one by a union bound argument and one using reduction from multiple source domains to single source domain. We conclude this section with a discussion and comparison of our bounds with existing generalization bounds for multisource domain adaptation [8, 35]. We refer readers to appendix for proof details and we mainly focus on discussing the interpretations and implications of the theorems.

Let $\{\mathcal{D}_{S_i}\}_{i=1}^k$ and $\mathcal{D}_T$ be $k$ source domains and the target domain, respectively. One idea to obtain a generalization bound for multiple source domains is to apply Thm. 1 repeatedly $k$ times, followed by a union bound to combine them. Following this idea, we first obtain the following bound as a corollary of Thm. 1 in the setting of multiple source domains, serving as a baseline model:

**Corollary 1** (Worst case classification bound). Let $\mathcal{H}$ be a hypothesis class with $VCdim(\mathcal{H}) = d$. If $\widehat{\mathcal{D}}_T$ and $\{\widehat{\mathcal{D}}_{S_i}\}_{i=1}^k$ are the empirical distributions generated with $m$ i.i.d. samples from each domain, then, for $0 < \delta < 1$, with probability at least $1 - \delta$, for all $h \in \mathcal{H}$, we have:

$$\varepsilon_T(h) \leq \max_{i \in [k]}\left\{\widehat{\varepsilon}_{S_i}(h) + \frac{1}{2}d_{\mathcal{H}\Delta\mathcal{H}}(\widehat{\mathcal{D}}_T; \widehat{\mathcal{D}}_{S_i}) + \lambda_i\right\} + O\left(\sqrt{\frac{1}{m}\left(\log\frac{k}{\delta} + d\log\frac{m}{d}\right)}\right) \qquad (2)$$

where $\lambda_i$ is the combined risk of the optimal hypothesis on domains $S_i$ and $T$.

This bound is quite pessimistic, as it essentially is a worst case bound, where the generalization on the target only depends on the worst source domain. However, in many real-world scenarios, when the number of related source domains is large, a single irrelevant source domain may not hurt the generalization too much. Furthermore, in the case of multiple source domains, despite the possible discrepancy between the source domains and the target domain, effectively we have a labeled sample of size $km$, while the asymptotic convergence rate in Corollary. 1 is of $\tilde{O}(\sqrt{1/m})$. Hence naturally one interesting question to ask is: is it possible to have a generalization bound of finite sample rate $\tilde{O}(\sqrt{1/km})$? In what follows we present a strategy to achieve a generalization bound of rate $\tilde{O}(\sqrt{1/km})$. The idea of this strategy is a reduction using convex combination from multiple domains to single domain by combining all the labeled instances from $k$ domains to one.

**Theorem 2** (Average case classification bound). Let $\mathcal{H}$ be a hypothesis class with $VCdim(\mathcal{H}) = d$. If $\{\widehat{\mathcal{D}}_{S_i}\}_{i=1}^k$ are the empirical distributions generated with $m$ i.i.d. samples from each domain, and $\widehat{\mathcal{D}}_T$ is the empirical distribution on the target domain generated from $mk$ samples without labels, then, $\forall \alpha \in \mathbb{R}_+^k, \sum_{i \in [k]} \alpha_i = 1$, and for $0 < \delta < 1$, w.p.b. at least $1 - \delta$, for all $h \in \mathcal{H}$, we have:

$$\varepsilon_T(h) \leq \sum_{i \in [k]} \alpha_i\left(\widehat{\varepsilon}_{S_i}(h) + \frac{1}{2}d_{\mathcal{H}\Delta\mathcal{H}}(\widehat{\mathcal{D}}_T; \widehat{\mathcal{D}}_{S_i})\right) + \lambda_\alpha + O\left(\sqrt{\frac{1}{km}\left(\log\frac{1}{\delta} + d\log\frac{km}{d}\right)}\right) \quad (3)$$

where $\lambda_\alpha$ is the risk of the optimal hypothesis on the mixture source domain $\sum_{i \in [k]} \alpha_i S_i$ and $T$.

Different from Corollary 1, Thm. 2 requires $mk$ unlabeled instances from the target domain. This is a mild requirement since unlabeled data is cheap to collect. Roughly, the bound in Thm. 2 can be understood as an average case bound if we choose $\alpha_i = 1/k, \forall i \in [k]$. Note that a simple convex

combination by applying Thm. 1 $k$ times can only achieve finite sample rate of $\tilde{O}(\sqrt{1/m})$, while the one in (3) achieves $\tilde{O}(\sqrt{1/km})$. On the other hand, the constants $\max_{i\in[k]} \lambda_i$ (in Corollary 1) and $\lambda_\alpha$ (in Thm. 2) are generally not comparable. As a final note, although the proof works for any convex combination $\alpha_i$, in the next section we will describe a practical method so that we do not need to explicitly choose it. Thm. 2 upper bounds the generalization error for classification problems. Next we also provide generalization guarantee for regression problem, where instead of VC dimension, we use pseudo-dimension to characterize the structural complexity of the hypothesis class.

**Theorem 3** (Average case regression bound). Let $\mathcal{H}$ be a set of real-valued functions from $\mathcal{X}$ to $[0, 1]$[2] with $Pdim(\mathcal{H}) = d$. If $\{\widehat{\mathcal{D}}_{S_i}\}_{i=1}^k$ are the empirical distributions generated with $m$ i.i.d. samples from each domain, and $\widehat{\mathcal{D}}_T$ is the empirical distribution on the target domain generated from $mk$ samples without labels, then, $\forall \alpha \in \mathbb{R}_+^k$, $\sum_{i\in[k]} \alpha_i = 1$, and for $0 < \delta < 1$, with probability at least $1 - \delta$, for all $h \in \mathcal{H}$, we have:

$$\varepsilon_T(h) \leq \sum_{i\in[k]} \alpha_i \left( \widehat{\varepsilon}_{S_i}(h) + \frac{1}{2} d_{\bar{\mathcal{H}}}(\widehat{\mathcal{D}}_T; \widehat{\mathcal{D}}_{S_i}) \right) + \lambda_\alpha + O\left( \sqrt{\frac{1}{km}\left( \log\frac{1}{\delta} + d\log\frac{km}{d} \right)} \right) \quad (4)$$

where $\lambda_\alpha$ is the risk of the optimal hypothesis on the mixture source domain $\sum_{i\in[k]} \alpha_i S_i$ and $T$, and $\bar{\mathcal{H}} := \{\mathbb{I}_{|h(x)-h'(x)|>t} : h, h' \in \mathcal{H}, 0 \leq t \leq 1\}$ is the set of threshold functions induced from $\mathcal{H}$.

**Comparison with Existing Bounds**. First, it is easy to see that, the bounds in both (2) and (3) reduce to the one in Thm. 1 when there is only one source domain ($k = 1$). Blitzer et al. [9] give a generalization bound for semi-supervised classification with multiple sources where, besides labeled instances from multiple source domains, the algorithm also has access to a fraction of labeled instances from the target domain. Although in general our bound and the one in [9, Thm. 3] are incomparable, it is instructive to see the connections and differences between them: our bound works in the unsupervised domain adaptation setting where we do not have any labeled data from the target. As a comparison, their bound in [9, Thm. 3] is a bound for semi-supervised domain adaptation. As a result, because of the access to labeled instances from the target domain, their bound is expressed relative to the optimal error on the target, while ours is in terms of the empirical error on the source domains, hence theirs is more informative. To the best of our knowledge, our bound in Thm. 3 is the first one using the idea of $\mathcal{H}$-divergence for regression problems. The proof of this theorem relies on a reduction from regression to classification. Mansour et al. [34] give a generalization bound for multisource domain adaptation under the assumption that the target distribution is a mixture of the $k$ sources and the target hypothesis can be represented as a convex combination of the source hypotheses. Also, their generalized discrepancy measure can be applied for other loss functions.

## 4 Multisource Domain Adaptation with Adversarial Neural Networks

Motivated by the bounds given in the last section, in this section we propose our model, multisource domain adversarial networks (MDAN), with two versions: Hard version (as a baseline) and Soft version. Suppose we are given samples drawn from $k$ source domains $\{\mathcal{D}_{S_i}\}$, each of which contains $m$ instance-label pairs. Additionally, we also have access to unlabeled instances sampled from the target domain $\mathcal{D}_T$. Once we fix our hypothesis class $\mathcal{H}$, the last two terms in the generalization bounds (2) and (3) will be fixed; hence we can only hope to minimize the bound by minimizing the first two terms, i.e., the source training error and the discrepancy between source domains and target domain. The idea is to train a neural network to learn a representation with the following two properties: 1). indistinguishable between the $k$ source domains and the target domain; 2). informative enough for our desired task to succeed. Note that both requirements are necessary: without the second property, a neural network can learn trivial random noise representations for all the domains, and such representations cannot be distinguished by any discriminator; without the first property, the learned representation does not necessarily generalize to the unseen target domain.

One key observation that leads to a practical approximation of $d_{\mathcal{H}\Delta\mathcal{H}}(\widehat{\mathcal{D}}_T; \widehat{\mathcal{D}}_{S_i})$ from Ben-David et al. [7] is that computing the discrepancy measure is closely related to learning a classifier that is able to distinguish samples from different domains. Let $\widehat{\varepsilon}_{T,S_i}(h)$ be the empirical risk of hypothesis

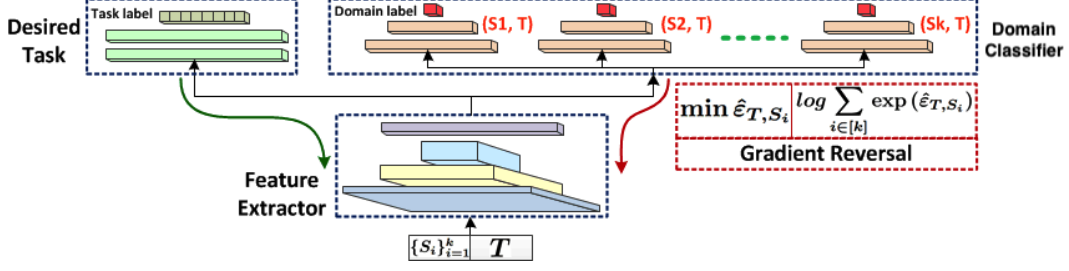

Figure 1: MDAN Network architecture. Feature extractor, domain classifier, and task learning are combined in one training process. Hard version: the source that achieves the minimum domain classification error is backpropagated with gradient reversal; Smooth version: all the domain classification risks over $k$ source domains are combined and backpropagated adaptively with gradient reversal.

$h$ in the domain discriminating task. Ignoring the constant terms that do not affect the upper bound, we can minimize the worst case upper bound in (2) by solving the following optimization problem:

$$\textbf{Hard version:} \quad \text{minimize} \quad \max_{i \in [k]} \left( \widehat{\varepsilon}_{S_i}(h) - \min_{h' \in \mathcal{H} \Delta \mathcal{H}} \widehat{\varepsilon}_{T, S_i}(h') \right) \tag{5}$$

The two terms in (5) exactly correspond to the two criteria we just proposed: the first term asks for an informative feature representation for our desired task to succeed, while the second term captures the notion of invariant feature representations between different domains. Inspired by Ganin et al. [17], we use the gradient reversal layer to effectively implement (5) by backpropagation. The network architecture is shown in Figure. 1. As discussed in the last section, one notable drawback of the hard version is that the algorithm may spend too much computational resources in optimizing the worst source domain. Furthermore, in each iteration the algorithm only updates its parameter based on the gradient from one of the $k$ domains. This is data inefficient and can waste our computational resources in the forward process.

To avoid both of the problems, we propose the MDAN Soft version that optimizes an upper bound of the convex combination bound given in (3). To this end, define $\widehat{\varepsilon}_i(h) := \widehat{\varepsilon}_{S_i}(h) - \min_{h' \in \mathcal{H} \Delta \mathcal{H}} \widehat{\varepsilon}_{T, S_i}(h')$ and let $\gamma > 0$ be a constant. We formulate the following optimization problem:

$$\textbf{Soft version:} \quad \text{minimize} \quad \frac{1}{\gamma} \log \sum_{i \in [k]} \exp \left( \gamma (\widehat{\varepsilon}_{S_i}(h) - \min_{h' \in \mathcal{H} \Delta \mathcal{H}} \widehat{\varepsilon}_{T, S_i}(h')) \right) \tag{6}$$

At the first glance, it may not be clear what the above objective function corresponds to. To understand this, if we define $\alpha_i = \exp(\widehat{\varepsilon}_i(h)) / \sum_{j \in [k]} \exp(\widehat{\varepsilon}_j(h))$, then the following inequality holds:

$$\sum_{i \in [k]} \alpha_i \widehat{\varepsilon}_i(h) \leq \log \left( \mathbb{E}_\alpha [\exp(\widehat{\varepsilon}_i(h))] \right) = \log \left( \frac{\sum_{i \in [k]} \exp^2(\widehat{\varepsilon}_i(h))}{\sum_{i \in [k]} \exp(\widehat{\varepsilon}_i(h))} \right) \leq \log \sum_{i \in [k]} \exp(\widehat{\varepsilon}_i(h))$$

In other words, the objective function in (6) is in fact an upper bound of the convex combination bound given in (3), with the combination weight $\alpha$ defined above. Compared with the one in (3), one advantage of the objective function in (6) is that we do not need to explicitly choose the value of $\alpha$. Instead, it adaptively corresponds to the loss $\widehat{\varepsilon}_i(h)$, and the larger the loss, the heavier the weight.

Alternatively, from the algorithmic perspective, during the optimization (6) naturally provides an adaptive weighting scheme for the $k$ source domains depending on their relative error. Use $\theta$ to denote all the model parameters:

$$\frac{\partial}{\partial \theta} \frac{1}{\gamma} \log \sum_{i \in [k]} \exp \left( \gamma (\widehat{\varepsilon}_{S_i}(h) - \min_{h' \in \mathcal{H} \Delta \mathcal{H}} \widehat{\varepsilon}_{T, S_i}(h')) \right) = \sum_{i \in [k]} \frac{\exp \gamma \widehat{\varepsilon}_i(h)}{\sum_{i' \in [k]} \exp \gamma \widehat{\varepsilon}_{i'}(h)} \frac{\partial \widehat{\varepsilon}_i(h)}{\partial \theta} \tag{7}$$

Compared with (5), the log-sum-exp trick not only smooths the objective, but also provides a principled and adaptive way to combine all the gradients from the $k$ source domains. In words, (7) says that the gradient of MDAN is a convex combination of the gradients from all the domains. The larger the error from one domain, the larger the combination weight in the ensemble. As we will see in Sec. 5, the optimization problem (6) often leads to better generalizations in practice, which may partly be explained by the ensemble effect of multiple sources implied by the upper bound.

# 5   Experiments

We evaluate both hard and soft MDAN and compare them with state-of-the-art methods on three
real-world datasets: the Amazon benchmark dataset [11] for sentiment analysis, a digit classification
task that includes 4 datasets: MNIST [29], MNIST-M [17], SVHN [37], and SynthDigits [17], and a
public, large-scale image dataset on vehicle counting from multiple city cameras [52]. Due to space
limit, details about network architecture and training parameters of proposed and baseline methods,
and detailed dataset description are described in appendix.

## 5.1   Amazon Reviews

Domains within the dataset consist of reviews on a specific kind of product (Books, DVDs, Electronics,
and Kitchen appliances). Reviews are encoded as 5000 dimensional feature vectors of unigrams and
bigrams, with binary labels indicating sentiment. We conduct 4 experiments: for each of them, we
pick one product as target domain and the rest as source domains. Each source domain has 2000
labeled examples, and the target test set has 3000 to 6000 examples. During training, we randomly
sample the same number of unlabeled target examples as the source examples in each mini-batch.
We implement both the Hard-Max and Soft-Max methods, and compare them with three baselines:
MLPNet, marginalized stacked denoising autoencoders (mSDA) [11], and DANN [17]. DANN
cannot be directly applied in multiple source domains setting. In order to make a comparison, we
use two protocols. The first one is to combine all the source domains into a single one and train
it using DANN, which we denote as C-DANN. The second protocol is to train multiple DANNs
separately, where each one corresponds to a source-target pair. Among all the DANNs, we report the
one achieving the best performance on the target domain. We denote this experiment as B-DANN.
For fair comparison, all these models are built on the same basic network structure with one input
layer (5000 units) and three hidden layers (1000, 500, 100 units).

Table 1: Sentiment classification accuracy.

| Train/Test | MLPNet | mSDA | B-DANN | C-DANN | MDAN | |
| --- | --- | --- | --- | --- | --- | --- |
| | | | | | Hard-Max | Soft-Max |
| **D+E+K/B** | 0.7655 | 0.7698 | 0.7650 | 0.7789 | 0.7845 | **0.7863** |
| **B+E+K/D** | 0.7588 | 0.7861 | 0.7732 | 0.7886 | 0.7797 | **0.8065** |
| **B+D+K/E** | 0.8460 | 0.8198 | 0.8381 | 0.8491 | 0.8483 | **0.8534** |
| **B+D+E/K** | 0.8545 | 0.8426 | 0.8433 | **0.8639** | 0.8580 | 0.8626 |

**Results and Analysis**. We show the accuracy of different methods in Table 1. Clearly, Soft-Max
significantly outperforms all other methods in most settings. When Kitchen is the target domain,
C-DANN performs slightly better than Soft-Max, and all the methods perform close to each other.
Hard-Max is typically slightly worse than Soft-Max. This is mainly due to the low data-efficiency
of the Hard-Max model (Section 4, Eq. 5, Eq. 6). We observe that with more training iterations,
the performance of Hard-Max can be further improved. These results verify the effectiveness of
MDAN for multisource domain adaptation. To validate the statistical significance of the results, we
also run a non-parametric Wilcoxon signed-ranked test for each task to compare Soft-Max with the
other competitors (see more details in appendix). From the statistical test, we see that Soft-Max is
convincingly better than other methods.

## 5.2   Digits Datasets

Following the setting in [17], we combine four digits datasets (MNIST, MNIST-M, SVHN, SynthDig-
its) to build the multisource domain dataset. We take each of MNIST-M, SVHN, and MNIST as
target domain in turn, and the rest as sources. Each source domain has $20,000$ labeled images and
the target test set has $9,000$ examples.

**Baselines**. We compare Hard-Max and Soft-Max of MDAN with 10 baselines: i). *B-Source*. A basic
network trained on each source domain ($20,000$ images) without domain adaptation and tested on
the target domain. Among the three models, we report the one achieves the best performance on the
test set. ii). *C-Source*. A basic network trained on a combination of three source domains ($20,000$
images for each) without domain adaptation and tested on the target domain. iii). *B-DANN*. We

Table 2: Accuracy on digit classification. T: MNIST; M: MNIST-M, S: SVHN, D: SynthDigits.

| Method | S+M+D/T | T+S+D/M | M+T+D/S | Method | S+M+D/T | T+S+D/M | M+T+D/S |
|--------|---------|---------|---------|--------|---------|---------|---------|
| **B-Source** | 0.964 | 0.519 | 0.814 | **C-Source** | 0.938 | 0.561 | 0.771 |
| **B-DANN** | 0.967 | 0.591 | **0.818** | **C-DANN** | 0.925 | 0.651 | 0.776 |
| **B-ADDA** | 0.968 | 0.657 | 0.800 | **C-ADDA** | 0.927 | 0.682 | 0.804 |
| **B-MTAE** | 0.862 | 0.534 | 0.703 | **C-MTAE** | 0.821 | 0.596 | 0.701 |
| **Hard-Max** | 0.976 | 0.663 | 0.802 | **Soft-Max** | **0.979** | **0.687** | 0.816 |
| **MDAC** | 0.755 | 0.563 | 0.604 | **Target** | 0.987 | 0.901 | 0.898 |

train DANNs [17] on each source-target domain pair (20, 000 images for each source) and test it on target. Again, we report the best score among the three. iv). *C-DANN*. We train a single DANN on a combination of three source domains (20, 000 images for each). v). *B-ADDA*. We train ADDA [46] on each source-target domain pair (20, 000 images for each source) and test it on the target domain. We report the best accuracy among the three. vi).*C-ADDA*. We train ADDA on a combination of three source domains (20, 000 images for each). vii). *B-MTAE*. We train MTAE [19] on each source-target domain pair (20, 000 images for each source) and test it on the target domain. We report the best accuracy among the three. viii). *C-MTAE*. We train MTAE on a combination of three source domains (20, 000 images for each). ix). *MDAC*. MDAC [51] is a multiple source domain adaptation algorithm that explores causal models to represent the relationship between the features $X$ and class label $Y$. We directly train MDAC on a combination of three source domains. x). *Target*. It is the basic network trained and tested on the target data. It serves as an upper bound of DA algorithms. All the MDAN and baseline methods are built on the same basic network structure to put them on a equal footing.

**Results and Analysis**. The classification accuracy is shown in Table 2. The results show that MDAN outperforms all the baselines in the first two experiments and is comparable with Best-Single-DANN in the third experiment. For the combined sources, MDAN always perform better than the source-only baseline (MDAN vs. Combine-Source). However, a naive combination of different training datasets can sometimes even decrease the performance of the baseline methods. This conclusion comes from three observations: First, directly training DANN on a combination of multiple sources leads to worse results than the source-only baseline (Combine-DANN vs. Combine-Source); Second, The performance of Combine-DANN can be even worse than the Best-Single-DANN (the first and third experiments); Third, directly training DANN on a combination of multiple sources always has lower accuracy compared with our approach (Combine-DANN vs. MDAN). We have similar observations for ADDA and MTAE. Such observations verify that the domain adaptation methods designed for single source lead to suboptimal solutions when applied to multiple sources. It also verifies the necessity and superiority of MDAN for multiple source adaptation. Furthermore, we observe that adaptation to the SVHN dataset (the third experiment) is hard. In this case, increasing the number of source domains does not help. We conjecture this is due to the large dissimilarity between the SVHN data to the others. Surprisingly, using a single domain (best-Single DANN) in this case achieves the best result. This indicates that in domain adaptation the quality of data (how close to the target data) is much more important than the quantity (how many source domains). As a conclusion, this experiment further demonstrates the effectiveness of MDAN when there are multiple source domains available, where a naive combination of multiple sources using DANN may hurt generalization.

## 5.3 WebCamT Vehicle Counting Dataset

WebCamT is a public dataset for vehicle counting from large-scale city camera videos, which has low resolution ($352 \times 240$), low frame rate (1 frame/second), and high occlusion. It has 60, 000 frames annotated with vehicle bounding box and count, divided into training and testing sets, with 42, 200 and 17, 800 frames, respectively. Here we demonstrate the effectiveness of MDAN to count vehicles from an unlabeled target camera by adapting from multiple labeled source cameras: we select 8 cameras located in different intersections of the city with different scenes, and each has more than 2, 000 labeled images for our evaluations. Among these 8 cameras, we randomly pick two cameras and take each camera as the target camera, with the other 7 cameras as sources. We compute the proxy $\mathcal{A}$-distance (PAD) [7] between each source camera and the target camera to approximate the divergence between them. We then rank the source cameras by the PAD from low to high and choose the first $k$ cameras to form the $k$ source domains. Thus the proposed methods and baselines can be evaluated on different numbers of sources (from 2 to 7). We implement the Hard-Max and Soft-Max MDAN, based on the basic vehicle counting network FCN [52]. We compare our method with two baselines: FCN [52], a basic network without domain adaptation, and DANN [17], implemented

Table 3: Counting error statistics. S is the number of source cameras; T is the target camera id.

| S | T | MDAN | | DANN | FCN | T | MDAN | | DANN | FCN |
|---|---|---|---|---|---|---|---|---|---|---|
| | | Hard-Max | Soft-Max | | | | Hard-Max | Soft-Max | | |
| 2 | A | 1.8101 | **1.7140** | 1.9490 | 1.9094 | B | 2.5059 | **2.3438** | 2.5218 | 2.6528 |
| 3 | A | 1.3276 | **1.2363** | 1.3683 | 1.5545 | B | 1.9092 | **1.8680** | 2.0122 | 2.4319 |
| 4 | A | 1.3868 | **1.1965** | 1.5520 | 1.5499 | B | **1.7375** | 1.8487 | 2.1856 | 2.2351 |
| 5 | A | 1.4021 | **1.1942** | 1.4156 | 1.7925 | B | 1.7758 | **1.6016** | 1.7228 | 2.0504 |
| 6 | A | 1.4359 | **1.2877** | 2.0298 | 1.7505 | B | 1.5912 | **1.4644** | 1.5484 | 2.2832 |
| 7 | A | 1.4381 | **1.2984** | 1.5426 | 1.7646 | B | 1.5989 | **1.5126** | 1.5397 | 1.7324 |

on top of the same basic network. We record mean absolute error (MAE) between true count and estimated count.

**Results and Analysis**. The counting error of different methods is compared in Table 3. The Hard-Max version achieves lower error than DANN and FCN in most settings for both target cameras. The Soft-Max approximation outperforms all the baselines and the Hard-Max in most settings, demonstrating the effectiveness of the smooth and adaptative approximation. The lowest MAE achieved by Soft-Max is $1.1942$. Such MAE means that there is only around one vehicle miscount for each frame (the average number of vehicles in one frame is around 20). Fig. 2 shows the counting results of Soft-Max for the two target cameras under the 5 source cameras setting. We can see that the proposed method accurately counts the vehicles of each target camera for long time sequences. Does adding more source cameras always help improve the performance on the target camera? To answer this question, we analyze the counting error when we vary the number of source cameras as shown in Fig. 3a, where the $x$-axis refers to number of source cameras and the $y$-axis includes both the MAE curve on the target camera as well as the PAD distance (bar chart) between the pair of source and target cameras. From the curves, we see the counting error goes down with more source cameras at the beginning, while it goes up when more sources are added at the end. This phenomenon shows that the performance on the target domain also depends on the its distance to the added source domain, i.e., it is not always beneficial to naively incorporate more source domains into training. To illustrate this better, we also show the PAD of the newly added camera in the bar chart of Fig. 3a. By observing the PAD and the counting error, we see the performance on the target can degrade when the newly added source camera has large divergence from the target camera. To show that MDAN can indeed decrease the divergences between target domain and multiple source domains, in Fig. 3b we plot the PAD distances between the target domains and the corresponding source domains. We can see that MDAN consistently decrease the PAD distances between all pairs of target and source domains, for both camera A and camera B. From this experiment we conclude that our proposed MDAN models are effective in multiple source domain adaptation.

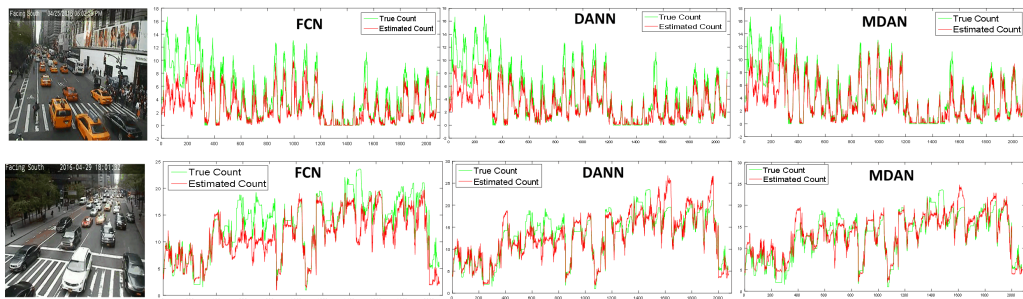

Figure 2: Counting results for target camera A (first row) and B (second row). X-frames; Y-Counts.

## 6   Related Work

A number of adaptation approaches have been studied in recent years. From the theoretical aspect, several theoretical results have been derived in the form of upper bounds on the generalization target error by learning from the source data. A keypoint of the theoretical frameworks is estimating the distribution shift between source and target. Kifer et al. [27] proposed the $\mathcal{H}$-divergence to

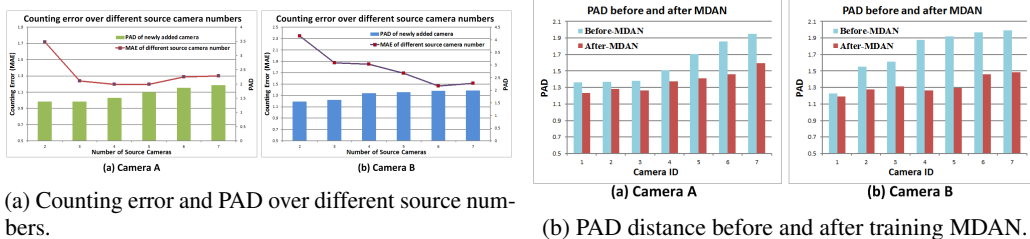

(a) Counting error and PAD over different source numbers.

(b) PAD distance before and after training MDAN.

Figure 3: PAD distance over different source domains along with their changes before and after training MDAN.

measure the similarity between two domains and derived a generalization bound on the target domain using empirical error on the source domain and the $\mathcal{H}$-divergence between the source and the target. This idea has later been extended to multisource domain adaptation [9] and the corresponding generalization bound has been developed as well. Ben-David et al. [8] provide a generalization bound for domain adaptation on the target risk which generalizes the standard bound on the source risk. This work formalizes a natural intuition of DA: reducing the two distributions while ensuring a low error on the source domain and justifies many DA algorithms. Based on this work, Mansour et al. [33] introduce a new divergence measure: discrepancy distance, whose empirical estimate is based on the Rademacher complexity [28]. See [13, 35] for more details.

Following the theoretical developments, many DA algorithms have been proposed, such as instance-based methods [44]; feature-based methods [6]; and parameter-based methods [15]. Recent studies have shown that deep neural networks can learn more transferable features for DA [14, 20, 50]. Bousmalis et al. [10] develop domain separation networks to extract image representations that are partitioned into two subspaces: domain private component and cross-domain shared component. The partitioned representation is utilized to reconstruct the images from both domains, improving the DA performance. Ganin et al. [17] propose a domain-adversarial neural network to learn the domain indiscriminate but main-task discriminative features. Adversarial training techniques that aim to build feature representations that are indistinguishable between source and target domains have been proposed in the last few years [2, 17]. Specifically, one of the central ideas is to use neural networks, which are powerful function approximators, to approximate a distance measure known as the $\mathcal{H}$-divergence between two domains [7, 8, 27]. The overall algorithm can be viewed as a zero-sum two-player game: one network tries to learn feature representations that can fool the other network, whose goal is to distinguish representations generated from the source domain between those generated from the target domain. The goal of the algorithm is to find a Nash-equilibrium of the game. Ideally, at such equilibrium state, feature representations from the source domain will share the same distributions as those from the target domain. Although these works generally outperform non-deep learning based methods, they only focus on the single-source-single-target DA problem, and much work is rather empirical design without statistical guarantees. Hoffman et al. [23] present a domain transform mixture model for multisource DA, which is based on non-deep architectures and is difficult to scale up.

# 7 Conclusion

We theoretically analyze generalization bounds for DA under the setting of multiple source domains with labeled instances and one target domain with unlabeled instances. Specifically, we propose average case generalization bounds for both classification and regression problems. The new bounds have interesting interpretations and the one for classification reduces to an existing bound when there is only one source domain. Following our theoretical results, we propose two MDAN to learn feature representations that are invariant under multiple domain shifts while at the same time being discriminative for the learning task. Both hard and soft versions of MDAN are generalizations of the popular DANN to the case when multiple source domains are available. Empirically, MDAN outperforms the state-of-the-art DA methods on three real-world datasets, including a sentiment analysis task, a digit classification task, and a visual vehicle counting task, demonstrating its effectiveness in multisource domain adaptation for both classification and regression problems.

**Acknowledgement**

HZ and GG gratefully acknowledge support from ONR, award number N000141512365. This research was supported in part by Fundação para a Ciência e a Tecnologia (project FCT [SFRH/BD/113729/2015] and a grant from the Carnegie Mellon-Portugal program). HZ would also like to thank Remi Tachet for his valuable comments and suggestions.

## Footnotes

[2]This is just for the simplicity of presentation, the range can easily be generalized to any bounded set.

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
