[Supplementary Material]

# A  Appendix

Organization of the appendix: 1). We first list the pseudocode of the proposed algorithms in Sec. B. 2). We then provide detailed proofs for all the corollaries and theorems presented in the main paper in Sec. C. 3). We describe more experimental details in Sec. D, including dataset description, network architecture and training parameters of the proposed and baseline methods, and more analysis of the experimental results. 4). We introduce and discuss more related work about domain adaptation in Sec. E.

# B  Pseudocode

Due to space limit of the main paper, we provide pseudocode of both algorithms, including the hard version and the soft version, in this section.

---

**Algorithm 1** Multiple Source Domain Adaptation

---

1: **for** $t = 1$ to $\infty$ **do**
2:     Sample $\{S_i^{(t)}\}_{i=1}^k$ and $T^{(t)}$ from $\{\widehat{\mathcal{D}}_{S_i}\}_{i=1}^k$ and $\widehat{\mathcal{D}}_T$, each of size $m$
3:     **for** $i = 1$ to $k$ **do**
4:         $\widehat{\varepsilon}_i^{(t)} \leftarrow \widehat{\varepsilon}_{S_i^{(t)}}(h) - \min_{h' \in \mathcal{H} \Delta \mathcal{H}} \widehat{\varepsilon}_{T^{(t)}, S_i^{(t)}}(h')$
5:         Compute $w_i^{(t)} := \exp(\widehat{\varepsilon}_i^{(t)})$
6:     **end for**
7:     Select $i^{(t)} := \arg\max_{i \in [k]} \widehat{\varepsilon}_i^{(t)}$                  // Hard version
8:     Backpropagate gradient of $\widehat{\varepsilon}_{i^{(t)}}^{(t)}$
9:     **for** $i = 1$ to $k$ **do**
10:        Normalize $w_i^{(t)} \leftarrow w_i^{(t)} / \sum_{i' \in [k]} w_{i'}^{(t)}$         // Soft version
11:     **end for**
12:     Backpropagate gradient of $\sum_{i \in [k]} w_i^{(t)} \widehat{\varepsilon}_i^{(t)}$
13: **end for**

---

# C  Proofs

## C.1  Proof of Corollary 1

**Corollary 1** (Worst case classification bound). Let $\mathcal{H}$ be a hypothesis class with $VCdim(\mathcal{H}) = d$. If $\widehat{\mathcal{D}}_T$ and $\{\widehat{\mathcal{D}}_{S_i}\}_{i=1}^k$ are the empirical distributions generated with $m$ i.i.d. samples from each domain, then, for $0 < \delta < 1$, with probability at least $1 - \delta$, for all $h \in \mathcal{H}$, we have:

$$\varepsilon_T(h) \leq \max_{i \in [k]} \left\{ \widehat{\varepsilon}_{S_i}(h) + \frac{1}{2} d_{\mathcal{H} \Delta \mathcal{H}}(\widehat{\mathcal{D}}_T; \widehat{\mathcal{D}}_{S_i}) + \lambda_i \right\} + O\left( \sqrt{\frac{1}{m} \left( \log \frac{k}{\delta} + d \log \frac{m}{d} \right)} \right) \quad (2)$$

*Proof.* For each one of the $k$ source domain, from Thm. 1, for $\forall \delta > 0$, w.p.b $\geq 1 - \delta/k$, we have the following inequality hold:

$$\varepsilon_T(h) \leq \widehat{\varepsilon}_{S_i}(h) + \frac{1}{2} d_{\mathcal{H} \Delta \mathcal{H}}(\widehat{\mathcal{D}}_{S_i}, \widehat{\mathcal{D}}_T) + \lambda_i + O\left( \sqrt{\frac{d \log(m/d) + \log(k/\delta)}{m}} \right)$$

Using a union bound argument, we have:

$$\Pr\left(\varepsilon_T(h) > \max_{i\in[k]}\left\{\widehat{\varepsilon}_{S_i}(h) + \frac{1}{2}d_{\mathcal{H}\Delta\mathcal{H}}(\widehat{\mathcal{D}}_T; \widehat{\mathcal{D}}_{S_i}) + \lambda_i\right\} + O\left(\sqrt{\frac{1}{m}\left(\log\frac{k}{\delta} + d\log\frac{m}{d}\right)}\right)\right)$$

$$\leq \Pr\left(\bigvee_{i\in[k]} \varepsilon_T(h) > \widehat{\varepsilon}_{S_i}(h) + \frac{1}{2}d_{\mathcal{H}\Delta\mathcal{H}}(\widehat{\mathcal{D}}_T; \widehat{\mathcal{D}}_{S_i}) + \lambda_i + O\left(\sqrt{\frac{1}{m}\left(\log\frac{k}{\delta} + d\log\frac{m}{d}\right)}\right)\right)$$

$$\leq \sum_{i\in[k]} \Pr\left(\varepsilon_T(h) > \widehat{\varepsilon}_{S_i}(h) + \frac{1}{2}d_{\mathcal{H}\Delta\mathcal{H}}(\widehat{\mathcal{D}}_T; \widehat{\mathcal{D}}_{S_i}) + \lambda_i + O\left(\sqrt{\frac{1}{m}\left(\log\frac{k}{\delta} + d\log\frac{m}{d}\right)}\right)\right)$$

$$\leq \sum_{i\in[k]} \delta/k = \delta$$

which completes the proof. ∎

## C.2  Proof of Theorem 2

**Theorem 2** (Average case classification bound). *Let $\mathcal{H}$ be a hypothesis class with $VCdim(\mathcal{H}) = d$. If $\{\widehat{\mathcal{D}}_{S_i}\}_{i=1}^k$ are the empirical distributions generated with $m$ i.i.d. samples from each domain, and $\widehat{\mathcal{D}}_T$ is the empirical distribution on the target domain generated from $mk$ samples without labels, then, $\forall\alpha\in\mathbb{R}_+^k, \sum_{i\in[k]}\alpha_i = 1$, and for $0 < \delta < 1$, w.p.b. at least $1 - \delta$, for all $h\in\mathcal{H}$, we have:*

$$\varepsilon_T(h) \leq \sum_{i\in[k]} \alpha_i\left(\widehat{\varepsilon}_{S_i}(h) + \frac{1}{2}d_{\mathcal{H}\Delta\mathcal{H}}(\widehat{\mathcal{D}}_T; \widehat{\mathcal{D}}_{S_i})\right) + \lambda_\alpha + O\left(\sqrt{\frac{1}{km}\left(\log\frac{1}{\delta} + d\log\frac{km}{d}\right)}\right) \quad (3)$$

*where $\lambda_\alpha$ is the risk of the optimal hypothesis on the mixture source domain $\sum_{i\in[k]}\alpha_i S_i$ and $T$.*

*Proof.* Consider a mixture distribution of the $k$ source domains where the mixture weight is given by $\alpha$. Denote it as $\mathcal{D}_{\tilde{S}} := \sum_{i\in[k]}\alpha_i\mathcal{D}_{S_i}$. Let $\tilde{S}$ be the combined samples from $k$ domains, then equivalently $\tilde{S}$ can be seen as a sample of size $km$ sampled i.i.d. from $\mathcal{D}_{\tilde{S}}$. Apply Thm. 1 using $\mathcal{D}_T$ as the target domain and $\mathcal{D}_{\tilde{S}}$ as the source domain, we know that for $0 < \delta < 1$, w.p.b. at least $1 - \delta$,

$$\varepsilon_T(h) \leq \widehat{\varepsilon}_{\tilde{S}}(h) + \frac{1}{2}d_{\mathcal{H}\Delta\mathcal{H}}(\widehat{\mathcal{D}}_{\tilde{S}}, \widehat{\mathcal{D}}_T) + \lambda_\alpha + O\left(\sqrt{\frac{d\log(km/d) + \log(1/\delta)}{km}}\right) \quad (8)$$

On the other hand, for $\forall h\in\mathcal{H}$, we have:

$$\widehat{\varepsilon}_{\tilde{S}}(h) = \sum_{i\in[k]}\alpha_i\widehat{\varepsilon}_{S_i}(h)$$

and we can upper bound $d_{\mathcal{H}\Delta\mathcal{H}}(\widehat{\mathcal{D}}_{\tilde{S}}, \widehat{\mathcal{D}}_T)$ as follows:

$$d_{\mathcal{H}\Delta\mathcal{H}}(\widehat{\mathcal{D}}_{\tilde{S}}, \widehat{\mathcal{D}}_T) = 2\sup_{A\in\mathcal{A}_{\mathcal{H}\Delta\mathcal{H}}}|\Pr_{\widehat{\mathcal{D}}_{\tilde{S}}}(A) - \Pr_{\widehat{\mathcal{D}}_T}(A)|$$

$$= 2\sup_{A\in\mathcal{A}_{\mathcal{H}\Delta\mathcal{H}}}|\sum_{i\in[k]}\alpha_i(\Pr_{\widehat{\mathcal{D}}_{S_i}}(A) - \Pr_{\widehat{\mathcal{D}}_T}(A))|$$

$$\leq 2\sup_{A\in\mathcal{A}_{\mathcal{H}\Delta\mathcal{H}}}\sum_{i\in[k]}\alpha_i|\Pr_{\widehat{\mathcal{D}}_{S_i}}(A) - \Pr_{\widehat{\mathcal{D}}_T}(A)|$$

$$\leq 2\sum_{i\in[k]}\alpha_i\sup_{A\in\mathcal{A}_{\mathcal{H}\Delta\mathcal{H}}}|\Pr_{\widehat{\mathcal{D}}_{S_i}}(A) - \Pr_{\widehat{\mathcal{D}}_T}(A)|$$

$$= \sum_{i\in[k]}\alpha_i d_{\mathcal{H}\Delta\mathcal{H}}(\widehat{\mathcal{D}}_{S_i}, \widehat{\mathcal{D}}_T)$$

where the first inequality is due to the triangle inequality and the second inequality is by the sub-additivity of the sup function. Replace $\widehat{\varepsilon}_{\tilde{S}}(h)$ with $\sum_{i\in[k]}\alpha_i\widehat{\varepsilon}_{S_i}(h)$ and upper bound $d_{\mathcal{H}\Delta\mathcal{H}}(\widehat{\mathcal{D}}_{\tilde{S}}, \widehat{\mathcal{D}}_T)$ by $\sum_{i\in[k]}\alpha_i d_{\mathcal{H}\Delta\mathcal{H}}(\widehat{\mathcal{D}}_{S_i}, \widehat{\mathcal{D}}_T)$ in (8) completes the proof. ∎

## C.3 Proof of Theorem 3

Before we give a full proof, we first describe the proof strategy at a high level. Roughly, the proof contains three parts. The first part contains a reduction from regression to classification by relating the pseudo-dimension of the hypothesis class $\mathcal{H}$ and its corresponding threshold binary classifiers. The second part uses $\mathcal{H}$-divergence to relate the source and target domains when they differ. The last part uses standard generalization analysis with pseudo-dimension for regression, when the source and target domains coincide.

### C.3.1 First Part of the Proof

To begin with, let $\mathcal{H} = \{h : \mathcal{X} \to [0,1]\}$ be a set of bounded real-valued functions from the input space $\mathcal{X}$ to $[0,1]$. We use $Pdim(\mathcal{H})$ to denote the pseudo-dimension of $\mathcal{H}$, and let $Pdim(\mathcal{H}) = d$. We first prove the following lemma that will be used in proving the main theorem:

**Lemma 1.** For $h, h' \in \mathcal{H} := \{h : \mathcal{X} \to [0,1]\}$, where $Pdim(\mathcal{H}) = d$, and for any distribution $\mathcal{D}_S$, $\mathcal{D}_T$ over $\mathcal{X}$,

$$|\varepsilon_S(h, h') - \varepsilon_T(h, h')| \leq \frac{1}{2} d_{\bar{\mathcal{H}}}(\mathcal{D}_S, \mathcal{D}_T)$$

where $\bar{\mathcal{H}} := \{\mathbb{I}_{|h(x)-h'(x)|>t} : h, h' \in \mathcal{H}, 0 \leq t \leq 1\}$.

*Proof.* By definition, for $\forall h, h' \in \mathcal{H}$, we have:

$$|\varepsilon_S(h, h') - \varepsilon_T(h, h')| \leq \sup_{h,h' \in \mathcal{H}} |\varepsilon_S(h, h') - \varepsilon_T(h, h')|$$

$$= \sup_{h,h' \in \mathcal{H}} \left| \mathbb{E}_{\mathbf{x}\sim S}[|h(\mathbf{x}) - h'(\mathbf{x})|] - \mathbb{E}_{\mathbf{x}\sim T}[|h(\mathbf{x}) - h'(\mathbf{x})|] \right| \quad (9)$$

Since $||h||_\infty \leq 1, \forall h \in \mathcal{H}$, then $0 \leq |h(\mathbf{x}) - h'(\mathbf{x})| \leq 1, \forall \mathbf{x} \in \mathcal{X}, h, h' \in \mathcal{H}$. We now use Fubini's theorem to bound $\left| \mathbb{E}_{\mathbf{x}\sim S}[|h(\mathbf{x}) - h'(\mathbf{x})|] - \mathbb{E}_{\mathbf{x}\sim T}[|h(\mathbf{x}) - h'(\mathbf{x})|] \right|$:

$$\left| \mathbb{E}_{\mathbf{x}\sim S}[|h(\mathbf{x}) - h'(\mathbf{x})|] - \mathbb{E}_{\mathbf{x}\sim T}[|h(\mathbf{x}) - h'(\mathbf{x})|] \right|$$

$$= \left| \int_0^1 \left( \Pr_S(|h(\mathbf{x}) - h'(\mathbf{x})| > t) - \Pr_T(|h(\mathbf{x}) - h'(\mathbf{x})| > t) \right) dt \right|$$

$$\leq \int_0^1 \left| \Pr_S(|h(\mathbf{x}) - h'(\mathbf{x})| > t) - \Pr_T(|h(\mathbf{x}) - h'(\mathbf{x})| > t) \right| dt$$

$$\leq \sup_{t \in [0,1]} \left| \Pr_S(|h(\mathbf{x}) - h'(\mathbf{x})| > t) - \Pr_T(|h(\mathbf{x}) - h'(\mathbf{x})| > t) \right|$$

Now in view of (9) and $\bar{\mathcal{H}}$, we have:

$$\sup_{h,h' \in \mathcal{H}} \sup_{t \in [0,1]} \left| \Pr_S(|h(\mathbf{x}) - h'(\mathbf{x})| > t) - \Pr_T(|h(\mathbf{x}) - h'(\mathbf{x})| > t) \right|$$

$$= \sup_{\bar{h} \in \bar{\mathcal{H}}} |\Pr_S(\bar{h}(\mathbf{x}) = 1) - \Pr_T(\bar{h}(\mathbf{x}) = 1)|$$

$$= \sup_{A \in \mathcal{A}_{\bar{\mathcal{H}}}} |\Pr_S(A) - \Pr_T(A)|$$

$$= \frac{1}{2} d_{\bar{\mathcal{H}}}(\mathcal{D}_S, \mathcal{D}_T)$$

Combining all the inequalities above finishes the proof. ∎

Next we bound $Pdim(|\mathcal{H} - \mathcal{H}|)$:

**Lemma 2.** If $Pdim(\mathcal{H}) = d$, then $Pdim(|\mathcal{H} - \mathcal{H}|) \leq 2d$.

*Proof.* By the definition of pseudo-dimension, we immediately have $VCdim(\bar{\mathcal{H}}) = Pdim(|\mathcal{H}-\mathcal{H}|)$. Next observe that each function $g \in \bar{\mathcal{H}}$ could be represented as a two layer linear threshold neural network, with two hidden units, one bias input unit and one output unit. Specifically, since

$$|h(\mathbf{x}) - h'(\mathbf{x})| > t \iff \max\{h(\mathbf{x}) - h'(\mathbf{x}) - t, h'(\mathbf{x}) - h(\mathbf{x}) - t\} > 0$$

which is also equivalent to

$$\operatorname{sgn}(h(\mathbf{x}) - h'(\mathbf{x}) - t) + \operatorname{sgn}(h'(\mathbf{x}) - h(\mathbf{x}) - t) > 0 \tag{10}$$

The above expression can then be implemented by a two layer one output linear threshold neural network. Hence from [3, Chapter 6, 8], the VC dimension of $\bar{\mathcal{H}}$ is at most twice of the pseudo-dimension of $\mathcal{H}$, completing the proof. ∎

### C.3.2 Second Part of the Proof

One technical lemma we will frequently use is the triangular inequality w.r.t. $\varepsilon_{\mathcal{D}}(h)$, $\forall h \in \mathcal{H}$:

**Lemma 3.** For any hypothesis class $\mathcal{H}$ and any distribution $\mathcal{D}$ on $\mathcal{X}$, the following triangular inequality holds:

$$\forall h, h', f \in \mathcal{H}, \quad \varepsilon_{\mathcal{D}}(h, h') \leq \varepsilon_{\mathcal{D}}(h, f) + \varepsilon_{\mathcal{D}}(f, h')$$

*Proof.*

$$\varepsilon_{\mathcal{D}}(h, h') = \mathbb{E}_{\mathbf{x} \sim \mathcal{D}}[|h(\mathbf{x}) - h'(\mathbf{x})|] \leq \mathbb{E}_{\mathbf{x} \sim \mathcal{D}}[|h(\mathbf{x}) - f(\mathbf{x})| + |f(\mathbf{x}) - f(\mathbf{x})|] = \varepsilon_{\mathcal{D}}(h, f) + \varepsilon_{\mathcal{D}}(f, h')$$

∎

Now we prove the following lemma in the regression setting when there is only one source domain and one target domain.

**Lemma 4.** Let $\mathcal{H}$ be a set of real-valued function from $\mathcal{X}$ to $[0, 1]$, and $\mathcal{D}_S$, $\mathcal{D}_T$ be the source and target distributions, respectively. Define $\bar{\mathcal{H}} := \{\mathbb{I}_{|h(x) - h'(x)| > t} : h, h' \in \mathcal{H}, 0 \leq t \leq 1\}$. Then $\forall h \in \mathcal{H}$, the following inequality holds:

$$\varepsilon_T(h) \leq \varepsilon_S(h) + \frac{1}{2} d_{\bar{\mathcal{H}}}(\mathcal{D}_T; \mathcal{D}_S) + \lambda$$

where $\lambda := \inf_{h' \in \mathcal{H}} \varepsilon_S(h') + \varepsilon_T(h')$.

*Proof.* Let $h^* := \arg\min_{h' \in \mathcal{H}} \varepsilon_S(h') + \varepsilon_T(h')$. For $\forall h \in \mathcal{H}$:

$$\begin{aligned}
\varepsilon_T(h) &\leq \varepsilon_T(h^*) + \varepsilon_T(h, h^*) \\
&= \varepsilon_T(h^*) + \varepsilon_T(h, h^*) - \varepsilon_S(h, h^*) + \varepsilon_S(h, h^*) \\
&\leq \varepsilon_T(h^*) + |\varepsilon_T(h, h^*) - \varepsilon_S(h, h^*)| + \varepsilon_S(h, h^*) \\
&\leq \varepsilon_T(h^*) + \varepsilon_S(h, h^*) + \frac{1}{2} d_{\bar{\mathcal{H}}}(\mathcal{D}_T, \mathcal{D}_S) \\
&\leq \varepsilon_T(h^*) + \varepsilon_S(h) + \varepsilon_S(h^*) + \frac{1}{2} d_{\bar{\mathcal{H}}}(\mathcal{D}_T, \mathcal{D}_S) \\
&= \varepsilon_S(h) + \frac{1}{2} d_{\bar{\mathcal{H}}}(\mathcal{D}_T; \mathcal{D}_S) + \lambda
\end{aligned}$$

The first and fourth inequalities are by the triangle inequality, and the third one is from Lemma. 1. ∎

### C.3.3 Third Part of the Proof

In this part of the proof, we use concentration inequalities to bound the source domain error $\varepsilon_S(h)$ and the divergence $d_{\bar{\mathcal{H}}}(\mathcal{D}_T; \mathcal{D}_S)$ in Lemma 4.

We first introduce the following generalization theorem in the regression setting when the source and target distributions are the same:

**Lemma 5** (Thm. 10.6, [36]). Let $\mathcal{H}$ be a family of real-valued functions from $\mathcal{X}$ to $[0, 1]$. Assume that $Pdim(\mathcal{H}) = d$. Then, for $\forall \delta > 0$, w.p.b. at least $1 - \delta$ over the choice of a sample of size $m$, the following inequality holds for all $h \in \mathcal{H}$:

$$\varepsilon(h) \leq \hat{\varepsilon}(h) + O\left(\sqrt{\frac{1}{m}\left(\log\frac{1}{\delta} + d\log\frac{m}{d}\right)}\right)$$

The next lemma bounds the $\mathcal{H}$-divergence between the population distribution and its corresponding empirical distribution:

**Lemma 6.** Let $\mathcal{D}_S$ and $\mathcal{D}_T$ be the source and target distribution over $\mathcal{X}$, respectively. Let $\mathcal{H}$ be a class of real-valued functions from $\mathcal{X}$ to $[0, 1]$, with $Pdim(\mathcal{H}) = d$. Define $\bar{\mathcal{H}} := \{\mathbb{I}_{|h(x)-h'(x)|>t} : h, h' \in \mathcal{H}, 0 \le t \le 1\}$. If $S$ and $T$ are the empirical distributions of $\mathcal{D}_S$ and $\mathcal{D}_T$ generated with $m$ $i.i.d.$ samples from each domain, then, for $0 < \delta < 1$, w.p.b. at least $1 - \delta$, we have:

$$d_{\bar{\mathcal{H}}}(\mathcal{D}_S; \mathcal{D}_T) \le d_{\bar{\mathcal{H}}}(S; T) + O\left(\sqrt{\frac{1}{m}\left(\log\frac{1}{\delta} + d\log\frac{m}{d}\right)}\right)$$

*Proof.* By the definition of pseudo-dimension, we have $VCdim(\bar{\mathcal{H}}) = Pdim(|\mathcal{H} - \mathcal{H}|)$. Now from Lemma 2, $Pdim(|\mathcal{H} - \mathcal{H}|) \le 2Pdim(\mathcal{H}) = 2d$, so $VCdim(\bar{\mathcal{H}}) \le 2d$. The lemma then follows from Ben-David et al. [8, Lemma 1] using standard concentration inequality. ∎

### C.3.4 Proof of the Generalization Bound under Regression Setting

We first prove a generalization bound for regression problem when there is only one source domain and one target domain. The extension to multiple source domains follows exactly as the proof of Thm. 2.

**Theorem 4.** Let $\mathcal{H}$ be a set of real-valued function from $\mathcal{X}$ to $[0, 1]$ with $Pdim(\mathcal{H}) = d$. Let $\widehat{\mathcal{D}}_S$ ($\widehat{\mathcal{D}}_T$) be the empirical distribution induced by sample of size $m$ drawn from $\mathcal{D}_S$ ($\mathcal{D}_T$). Then w.p.b. at least $1 - \delta$, $\forall h \in \mathcal{H}$,

$$\varepsilon_T(h) \le \widehat{\varepsilon}_S(h) + \frac{1}{2}d_{\bar{\mathcal{H}}}(\widehat{\mathcal{D}}_S, \widehat{\mathcal{D}}_T) + \lambda + O\left(\sqrt{\frac{d\log(m/d) + \log(1/\delta)}{m}}\right) \qquad (11)$$

where $\lambda = \inf_{h' \in \mathcal{H}} \varepsilon_S(h') + \varepsilon_T(h')$ and $\bar{\mathcal{H}} := \{\mathbb{I}_{|h(x)-h'(x)|>t} : h, h' \in \mathcal{H}, 0 \le t \le 1\}$.

*Proof.* Combine Lemma 4, Lemma 5 and Lemma 6 using union bound finishes the proof. ∎

## D  Details about Experiments

In this section, we describe more details about the datasets and the experimental settings. We extensively evaluate the proposed methods on three datasets: 1). We first evaluate our methods on Amazon Reviews dataset [11] for sentiment analysis. 2). We evaluate the proposed methods on the digits classification datasets including MNIST [29], MNIST-M [17], SVHN [37], and SynthDigits [17]. 3). We further evaluate the proposed methods on the public dataset WebCamT [52] for vehicle counting. It contains 60,000 labeled images from 12 city cameras with different distributions. Due to the substantial difference between these datasets and their corresponding learning tasks, we will introduce more detailed dataset description, network architecture, and training parameters for each dataset respectively in the following subsections.

### D.1  Details on Amazon Reviews evaluation

Amazon reviews dataset includes four domains, each one composed of reviews on a specific kind of product (Books, DVDs, Electronics, and Kitchen appliances). Reviews are encoded as 5000 dimensional feature vectors of unigrams and bigrams. The labels are binary: 0 if the product is ranked up to 3 stars, and 1 if the product is ranked 4 or 5 stars.

We take one product domain as target and the other three as source domains. Each source domain has 2000 labeled examples and the target test set has 3000 to 6000 examples. We implement the Hard-Max and Soft-Max methods based on a basic network with one input layer (5000 units) and three hidden layers (1000, 500, 100 units). The network is trained for 50 epochs with dropout rate 0.7. We compare Hard-Max and Soft-Max with three baselines: *Baseline 1: MLPNet*. It is the basic network of our methods (one input layer and three hidden layers), trained for 50 epochs with dropout rate 0.01. *Baseline 2: Marginalized Stacked Denoising Autoencoders (mSDA)* [11]. It takes the unlabeled parts of both source and target samples to learn a feature map from input space to a

new representation space. As a denoising autoencoder algorithm, it finds a feature representation from which one can (approximately) reconstruct the original features of an example from its noisy counterpart. *Baseline 3: DANN*. We implement DANN based on the algorithm described in [17] with the same basic network as our methods. Hyper parameters of the proposed and baseline methods are selected by cross validation. Table 4 summarizes the network architecture and some hyper parameters.

Table 4: Network parameters for proposed and baseline methods

| Method | Input layer | Hidden layers | Epochs | Dropout | Domains | Adaptation weight | $\gamma$ |
|--------|-------------|---------------|--------|---------|---------|-------------------|----------|
| **MLPNet** | 5000 | (1000, 500, 100) | 50 | 0.01 | N/A | N/A | N/A |
| **DANN** | 5000 | (1000, 500, 100) | 50 | 0.01 | 1 | 0.01 | N/A |
| **MDAN** | 5000 | (1000, 500, 100) | 50 | 0.7 | 3 | 0.1 | 10 |

To validate the statistical significance of the results, we run a non-parametric Wilcoxon signed-ranked test for each task to compare Soft-Max with the other competitors, as shown in Table 5. Each cell corresponds to the $p$-value of a Wilcoxon test between Soft-Max and one of the other methods, under the null hypothesis that the two paired samples have the same mean. From these p-values, we see Soft-Max is convincingly better than other methods.

Table 5: $p$-values under Wilcoxon test.

| | MLPNet | mSDA | Best-Single-DANN | Combine-DANN | Hard-Max |
|---|--------|------|------------------|--------------|----------|
| | Soft-Max | Soft-Max | Soft-Max | Soft-Max | Soft-Max |
| **B** | 0.550 | 0.101 | 0.521 | 0.013 | 0.946 |
| **D** | 0.000 | 0.072 | 0.000 | 0.051 | 0.000 |
| **E** | 0.066 | 0.000 | 0.097 | 0.150 | 0.022 |
| **K** | 0.306 | 0.001 | 0.001 | 0.239 | 0.008 |

## D.2   Details on Digit Datasets evaluation

We evaluate the proposed methods on the digits classification problem. Following the experiments in [17], we combine four popular digits datasets-MNIST, MNIST-M, SVHN, and SynthDigits to build the multi-source domain dataset. MNIST is a handwritten digits database with $60,000$ training examples, and $10,000$ testing examples. The digits have been size-normalized and centered in a $28 \times 28$ image. MNIST-M is generated by blending digits from the original MNIST set over patches randomly extracted from color photos from BSDS500 [4, 17]. It has $59,001$ training images and $9,001$ testing images with $32 \times 32$ resolution. An output sample is produced by taking a patch from a photo and inverting its pixels at positions corresponding to the pixels of a digit. For DA problems, this domain is quite distinct from MNIST, for the background and the strokes are no longer constant. SVHN is a real-world house number dataset with $73,257$ training images and $26,032$ testing images. It can be seen as similar to MNIST, but comes from a significantly harder, unsolved, real world problem. SynthDigits consists of $500;000$ digit images generated by Ganin et al. [17] from WindowsTM fonts by varying the text, positioning, orientation, background and stroke colors, and the amount of blur. The degrees of variation were chosen to simulate SVHN, but the two datasets are still rather distinct, with the biggest difference being the structured clutter in the background of SVHN images.

We take MNIST-M, SVHN, and MNIST as target domain in turn, and the remaining three as sources. We implement the Hard-Max and Soft-Max versions according to Alg. 1 based on a basic network, as shown in Fig. 4. The baseline methods are also built on the same basic network structure to put them on a equal footing. The network structure and parameters of MDAN are illustrated in Fig. 4. The learning rate is initialized by $0.01$ and adjusted by the first and second order momentum in the training process. The domain adaptation parameter of MDAN is selected by cross validation. In each mini-batch of MDAN training process, we randomly sample the same number of unlabeled target images as the number of the source images.

Figure 4: MDAN network architecture for digit classification

## D.3 Details on WebCamT Vehicle Counting

WebCamT is a public dataset for large-scale city camera videos, which have low resolution ($352 \times 240$), low frame rate (1 frame/second), and high occlusion. WebCamT has $60,000$ frames annotated with rich information: bounding box, vehicle type, vehicle orientation, vehicle count, vehicle re-identification, and weather condition. The dataset is divided into training and testing sets, with 42,200 and 17,800 frames, respectively, covering multiple cameras and different weather conditions. WebCamT is an appropriate dataset to evaluate domain adaptation methods, for it covers multiple city cameras and each camera is located in different intersection of the city with different perspectives and scenes. Thus, each camera data has different distribution from others. The dataset is quite challenging and in high demand of domain adaptation solutions, as it has $6,000,000$ unlabeled images from 200 cameras with only $60,000$ labeled images from 12 cameras. The experiments on WebCamT provide an interesting application of our proposed MDAN: when dealing with spatially and temporally large-scale dataset with much variations, it is prohibitively expensive and time-consuming to label large amount of instances covering all the variations. As a result, only a limited portion of the dataset can be annotated, which can not cover all the data domains in the dataset. MDAN provide an effective solution for this kind of application by adapting the deep model from multiple source domains to the unlabeled target domain.

We evaluate the proposed methods on different numbers of source cameras. Each source camera provides 2000 labeled images for training and the test set has 2000 images from the target camera. In each mini-batch, we randomly sample the same number of unlabeled target images as the source images. We implement the Hard-Max and Soft-Max version of MDAN according to Alg. 1, based on the basic vehicle counting network FCN described in [52]. Please refer to [52] for detailed network architecture and parameters. The learning rate is initialized by $0.01$ and adjusted by the first and second order momentum in the training process. The domain adaptation parameter is selected by cross validation. We compare our method with two baselines: *Baseline 1: FCN*. It is our basic network without domain adaptation as introduced in work [52]. *Baseline 2: DANN*. We implement DANN on top of the same basic network following the algorithm introduced in work [17].

# E More Related Work

A number of adaptation approaches have been studied in recent years. From the theoretical aspect, several theoretical results have been derived in the form of upper bounds on the generalization target error by learning from the source data. A keypoint of the theoretical frameworks is estimating the distribution shift between source and target. Kifer et al. [27] proposed the $\mathcal{H}$-divergence to measure the similarity between two domains and derived a generalization bound on the target domain using empirical error on the source domain and the $\mathcal{H}$-divergence between the source and the target. This idea has later been extended to multisource domain adaptation [9] and the corresponding generalization bound has been developed as well. Ben-David et al. [8] provide a generalization bound for domain adaptation on the target risk which generalizes the standard bound on the source risk. This work formalizes a natural intuition of DA: reducing the two distributions while ensuring a low

Figure 5: Locations of the source&target camera map.

error on the source domain and justifies many DA algorithms. Based on this work, Mansour et al. [33] introduce a new divergence measure: discrepancy distance, whose empirical estimate is based on the Rademacher complexity [28] (rather than the VC-dim). Other theoretical works have also been studied such as [32] that derives the generalization bounds on the target error by taking use of the robustness properties introduced in [49]. See [13, 35] for more details.

Following the theoretical developments, many DA algorithms have been proposed, such as instance-based methods [44]; feature-based methods [6]; and parameter-based methods [15]. The general approach for domain adaptation starts from algorithms that focus on linear hypothesis class [12, 18]. The linear assumption can be relaxed and extended to the non-linear setting using the kernel trick, leading to a reweighting scheme that can be efficiently solved via quadratic programming [21]. Recently, due to the availability of rich data and powerful computational resources, non-linear representations and hypothesis classes have been increasingly explored [2, 5, 11, 17, 20]. This line of work focuses on building common and robust feature representations among multiple domains using either supervised neural networks [20], or unsupervised pretraining using denoising auto-encoders [47, 48].

Recent studies have shown that deep neural networks can learn more transferable features for DA [14, 20, 50]. Bousmalis et al. [10] develop domain separation networks to extract image representations that are partitioned into two subspaces: domain private component and cross-domain shared component. The partitioned representation is utilized to reconstruct the images from both domains, improving the DA performance. Reference [30] enables classifier adaptation by learning the residual function with reference to the target classifier. The main-task of this work is limited to the classification problem. Ganin et al. [17] propose a domain-adversarial neural network to learn the domain indiscriminate but main-task discriminative features. Although these works generally outperform non-deep learning based methods, they only focus on the single-source-single-target DA problem, and much work is rather empirical design without statistical guarantees. Hoffman et al. [23] present a domain transform mixture model for multisource DA, which is based on non-deep architectures and is difficult to scale up.

Adversarial training techniques that aim to build feature representations that are indistinguishable between source and target domains have been proposed in the last few years [2, 17]. Specifically, one of the central ideas is to use neural networks, which are powerful function approximators, to approximate a distance measure known as the $\mathcal{H}$-divergence between two domains [7, 8, 27]. The overall algorithm can be viewed as a zero-sum two-player game: one network tries to learn feature representations that can fool the other network, whose goal is to distinguish representations generated from the source domain between those generated from the target domain. The goal of the algorithm is to find a Nash-equilibrium of the game, or the stationary point of the min-max saddle point problem. Ideally, at such equilibrium state, feature representations from the source domain will share the same distributions as those from the target domain, and, as a result, better generalization on the target domain can be expected by training models using only labeled instances from the source domain.