[Reviews · NeurIPS 2018]

Reviewer 1



Quality: The paper tackles the problem of domain adaptation. The main proposal of the paper is to find a multi-source generalization bound equivalent of a previously established single-source bound in [9]. Accordingly, two objective functions are devised to tighten the bound for multi-source domain adaptation. The experiments clearly shows that optimizing the proposed objective is more effective than naive extension of single-source objectives. The main paper is technically sound both the theories and the experiments seem valid to the eye of the reviewer. Clarity: The paper is easy to follow. It motivates the problem, puts forward the formal notation in a concise and readable format and builds up the proposed approach on that. The hyper-parameters necessary to reproduce the experimental results come in the appendix. The paper would strongly benefit from a thorough theoretical and empirical analysis of the convex coefficients \alpha on the tightness of the bound as well as the optimization. The reviewer appreciates the design of intuitive adaptive coefficients in 6 but more discussions and analysis would have been very helpful. Originality: The paper builds on top of an existing work on single source domain adaptation and generalizes it to a special case of multi-source domain adaptation. The assumption about the setups of the multi-source domain adaptation (availability of mk unlabelled target data) is not too restrictive. The related works section is limited, other works on multi-source domain adaptation exist that are not cited. For instance, Sun et al. NIPS 2011, or Choi et al. CVPR 2018. Significance: - the paper puts forward a specific generalization of a previously published single-source domain adaptation error bound to multiple sources. The generalization is simple and intuitive. - The paper tests that optimizing the derived bounds is better than naive approaches to multi-source domain adaptation. In that regard, it conclusively shows the benefits. However, it fails to compare to other approaches that are taken for multi-source domain adaptation (such as the ones mentioned above). - Another issue with the experiments is that when comparing C-DANN with the proposed MDAN it seems like C-DANN can achieve comparable results and on rare occasions even better results. This raises the need for baselines where more advanced domain adaptation techniques are used with combined source domains as a single domain. ------------------- I read the rebuttal and the reviewers' comments. It seems I agree with all the reviewers' comments and the rebuttal makes sense. I will keep my rating and vouch for the acceptance of the paper.

Reviewer 2



This paper proposes a multisource domain adversarial network (MDAN) for unsupervised multi-source domain adaptation problem and claims the following contributions: 1) provides generalization bounds and efficient algorithms for both the classification and regression settings. Moreover, the authors empirically demonstrate that MDAN achieves superior performance over other state of the art approaches on multiple real-world datasets. Strong points: - This paper tackles a timely problem of unsupervised multi-source domain adaptation, and proposes an theoretically sound and elegant approach called MDAN to address it. MDAN combines feature extraction, domain classification, and task learning in one training process.Two versions of MDAN are presented - Hard version and Soft version - corresponding to the worst-case and average-case generalization bounds. The authors use H-divergence idea to provide bounds for regression problems- which is new. - The authors have conducted extensive experiments to showcase the performance of MDAN. The authors carefully explain the baselines, and analyze the results obtained. In the supplementary materials the authors provide pseudocode/algorithm, proofs, model architectures, and present Wilcoxon test results to showcase the statistical significance of the results. - This paper is well written and easy to read. Most of the recent works on the topic are discussed. Weak points: - There are a lot of recent works on unsupervised domain adaptation and authors compare their works to many relevant ones. However, some works such as Variational FAIR [1] which use a generative approach can have better results than DANN, and they are not cited or compared against. It is highly suggested that the authors either compare or discuss them. [1] Louizos, Christos, Kevin Swersky, Yujia Li, Max Welling, and Richard Zemel. "The variational fair autoencoder." ICLR (2016). - The generalization bounds appear as extensions of the single source-target pairs. - (minor) Couple of typos in the draft. Overall, I liked this paper as it provides good theoretical and empirical results on an important problem.

Reviewer 3



After reading the authors' response, I've updated the score accordingly. Now I'm clearer about the authors' intuition than before. My major concerns are: (1) d = O(p log p) as the VC dimension of a neural network is too large a number in the theorems, since the number of parameters in the neural network can reach millions easily. In consequence, it is saying that the target error is smaller than a super large number despite that d is within a reasonable range. (2) Though the results are not required to be SOTA, it should be at least comparable to SOTA. ~30% lower than SOTA still remains a problem. ___________________________________________________________________________ Overall: This work extends the previous work, called domain adaptation neural nets (DANN), to multiple sources. Clarity: The idea is clear. The writing is easy to follow. Originality: This work is incremental to previous research. It can be simplified by DANN with a multi-class domain classifier. For the proposed method MDAN, (i) the domain discriminator number is proportional to domain number, which is a problem in practice when the source domain number goes high. (ii) The proof is standard and straightforward using H-divergence. However, Corollary 1 and Theorem 2 requires the VC dimension of the encoder module. Since the a neural network is used, the VC dimension will go to ‘nearly’ infinity. It would be better if the authors can give some explanation on this point. (iii) The competitive methods are limited (only DANN and standard neural network) on the text classification and vehicle counting dataset. It would be better if the authors can compare with more recent works, like the kernel or optimal transport domain adaptation methods. For the digit image dataset, recent works can achieve much better results than the reported ones. For instance, Shu, Rui, et al. "A DIRT-T Approach to Unsupervised Domain Adaptation." ICLR 2018, reaches 98% on MNISTM dataset, which is 30% higher than the result given in table 2. Thus the comparison methods in the left part of table 2, which is best single source, should at least be more carefully selected. (iv) It would be better if the author can show the domain weights w in each setting. For instance, it is more likely that mnistm is matched to svhn since they are both colored images. However, mnist should contribute the most to mnistm since mnistm is generated from mnist. It would be interesting if the proposed method can solve this problem.